# An archaellum filament composed of two alternating subunits

Lavinia Gambelli [1,2,6], Michail N. Isupov[3,6], Rebecca Conners [1,4], Mathew McLaren[1,4], Annett Bellack [5], Vicki Gold [1,4], Reinhard Rachel [5] & Bertram Daum [1,4✉]

Archaea use a molecular machine, called the archaellum, to swim. The archaellum consists of an ATP-powered intracellular motor that drives the rotation of an extracellular filament composed of multiple copies of proteins named archaellins. In many species, several archaellin homologs are encoded in the same operon; however, previous structural studies indicated that archaellum filaments mainly consist of only one protein species. Here, we use electron cryo-microscopy to elucidate the structure of the archaellum from *Methanocaldococcus villosus* at 3.08 Å resolution. The filament is composed of two alternating archaellins, suggesting that the architecture and assembly of archaella is more complex than previously thought. Moreover, we identify structural elements that may contribute to the filament's flexibility.

[1] Living Systems Institute, University of Exeter, Exeter EX4 4QD, UK. [2] College of Engineering, Mathematics and Physical Sciences, University of Exeter, Exeter EX4 4QF, UK. [3] Henry Wellcome Building for Biocatalysis, Biosciences, College of Life and Environmental Sciences, University of Exeter, Exeter EX4 4QD, UK. [4] College of Life and Environmental Sciences, University of Exeter, Exeter EX4 4QD, UK. [5] Institute of Microbiology and Archaea Centre, University of Regensburg, 93053 Regensburg, Germany. [6] These authors contributed equally: Lavinia Gambelli, Michail N. Isupov. ✉email: b.daum2@exeter.ac.uk

Archaea are ubiquitous microorganisms that successfully colonise diverse environments partially due to their ability to swim by means of archaella[1–3]. The archaellum is a propulsive nanomachine consisting of an intracellular motor that drives the rotation of an extracellular filament[4]. A clockwise rotation of the archaellum moves the cell forward, whereas a counterclockwise rotation moves the cell backward[5]. While archaella, like bacterial flagella, mediate swimming, the two systems are not homologous, suggesting their independent evolution. Instead, archaella are structurally homologous to filaments of the type-IV filament (T4F) superfamily. The T4F superfamily includes a vast range of filamentous nanomachines, such as bacterial type-IV pili (T4P)[6], associated with diverse functions[7]. Within the T4F superfamily, archaella are unique in providing swimming motion by means of a rotary propeller. Other functions of T4F in archaea include surface adhesion[8–11], cell–cell contacts and biofilm formation[8,12].

The archaellum machinery consists of a membrane-embedded motor complex and a ~10 μm long filament, which together are formed by ~10 components, depending on the species[13]. The filament is a helical array of ArlA and/or ArlB archaellins[14] that assemble proximal to the cell surface[15]. Filament assembly and rotation is an ATP-dependent process driven by the motor complex, consisting of the platform protein ArlJ, the hexameric AAA + ATPase ArlI, and a putative regulator ArlH[6,16–18]. The motor is surrounded by a cytosolic ring of ArlX proteins (in Crenarchaeota)[19] or ArlC, D/E (in Euryarchaeota)[20,21]. In the periplasm, the motor complex is anchored to the S-layer by the ArlF and ArlG stator proteins[22,23].

Most genes encoding the archaellum machinery components are organised in one *arl* operon. The first genes in this operon (*arlA* or *arlB*) encode for the archaellin subunits that make up the archaellum filament. The number of archaellin genes is species-specific, usually varying from one to seven[24,25]. When multiple archaellins are encoded, several or all can be transcribed at the same time[26,27]. Conventionally, ArlA and ArlB archaellins are divided into major and minor structural components of the filament. Archaellins that are thought to form the bulk of the filament are referred to as major. Biochemical and genetic experiments suggest that in *Methanococcus voltae*[28], *Methanococcus maripaludis*[29], *Methanothermococcus thermolithotrophicus*[30], *Halobacterium salinarum*[31] and *Halorubrum lacusprofundi* DL18[32] the archaellum filament consists of two major archaellins, hypothesised to be equally distributed along the filament. Nevertheless, the three high-resolution structures of archaella that are available to date show homopolymeric filaments consisting of only a single (major) archaellin[20,33,34].

The roles of minor archaellins mostly remain enigmatic. Genetics and molecular biology experiments suggest that in *M. voltae*[35], *M. maripaludis*[29] and *H. salinarum*[36], minor archaellins form a region at the base of the archaellum filament that is reminiscent of the flagellar hook. However, not all archaea show this feature, including those that encode only one archaellin.

Here we employed electron cryo-microscopy (cryoEM) to solve the structure of the archaellum filament from *Methanocaldococcus villosus*, a hyperthermophilic archaeon[11] and one of the fastest swimming organisms known[37]. We report the structure of a heteropolymeric archaeal filament composed of two alternating subunits, ArlB1 and ArlB2 and provide insights into the dynamics of archaella. Our results provide a shift in our understanding of how archaella assemble and function.

## Results

**High-resolution cryoEM and helical reconstruction of *M. villosus* archaella.** The structure of the *M. villosus* archaellum was determined from cryoEM movies of isolated filaments using Relion 3.1[38] and cryoSPARC 3.1.0[39] (Supplementary Figs. 1, 2). Helical parameters from previously published archaella structures were applied and optimised by helical search to a rise of 5.57 Å and a twist of 108°. This approach yielded a 3.28 Å resolution map in which the filament appeared to consist of one archaellin homologue (Supplementary Fig. 3a). Surprisingly, further refinement of the cryoEM map without imposing helical symmetry produced two distinct and alternating protein densities (Fig. 1; Supplementary Fig. 4; Supplementary Movie 1; Supplementary Fig. 3b, c). Careful analysis of the filament's heteropolymeric structure revealed that the helical symmetry did not comply with the initially applied rise and twist. Instead, the minimal transformation that superimposes each archaellin subunit onto one of the same types is n + 6 (6 subunits along the filament long axis) (Supplementary Fig. 4e). Recalculating the helical symmetry of the archaellum based on this observation resulted in a helical rise of 33.4 Å (5.57 Å × 6 = 33.42 Å, refined to 33.4 Å) and a twist of −71.8° (108° × 6 = 648 = 720 − 72°). Refining our map with these parameters improved the resolution to 3.08 Å (Fig. 1; Supplementary Figs. 4 and 6).

**The *M. villosus* archaellum consists of the subunits ArlB1 and ArlB2.** The archaellum operon of *M. villosus* encodes three archaellin homologs (ArlB1, 2 and 3). To investigate which two constitute the filament, we built atomic models based on their sequences for both distinct protein densities. ArlB1 and ArlB2 could each be modelled unambiguously into alternating positions in the map, guided by aromatic side chains and glycosylation sites (Figs. 2 and 3; Supplementary Fig. 7; Supplementary Movie 2). The amino acid sequence of ArlB3, instead, did not match the map density features.

The proteinaceous part of the *M. villosus* archaellum filament has a ~9.8 nm diameter. ArlB1 and ArlB2 alternate throughout the filament, following left-handed helical strands. The archaellum is composed of three of these helical strands, which by convention are defined as 3-start helices (Figs. 1; 4a, b; Supplementary Fig. 8; Supplementary Movies 1 and 2).

ArlB1 and ArlB2 are homologous by structure and sequence (Fig. 2d–f; Supplementary Figs. 9, 10). Their genes encode proteins of 221 and 225 amino acids, respectively, which share 56.5% sequence identity (75.9% similarity). As is typical for other members of the T4F superfamily, ArlB1 and ArlB2 consist of an N-terminal hydrophobic α-helical "tail" and a globular, C-terminal β-strand rich "head" domain. In the assembled archaellum, the tails constitute the core of the filament, whereas the head domains are exposed to the periphery (Fig. 2a–c). The first 12 amino acids are not present in the mature ArlB1 and ArlB2 proteins due to N-terminal processing by a class-III signal peptidase prior to their insertion into the filament[40]. While the α-helical tail domains are identical in ArlB1 and ArlB2 ($I_{14}–A_{59}$), the head domains show differences with respect to sequence and structure (Fig. 2f; Supplementary Figs. 9, 10). ArlB1 consists of 12 β-strands and three α-helices, whereas ArlB2 has 13 β-strands and two α-helices. The most striking difference is a glycosylated outward-facing loop ($K_{131}–A_{137}$) in ArlB2, which is absent in ArlB1 (Figs. 2f and 3c).

**N-glycosylation and metal-binding sites.** Refinement of the helical parameters was crucial to clearly resolve the glycosylation pattern of the archaellum filament. Glycan densities were identified as dead-end protrusions from the peptide backbone that were larger than side chains and co-localised with the canonical N-glycosylation consensus sequon (N-X-S/T) (Fig. 3a–c, Supplementary Fig. 7). Including the density occupied by the glycan moieties, the diameter of the archaellum is ~11 nm. All

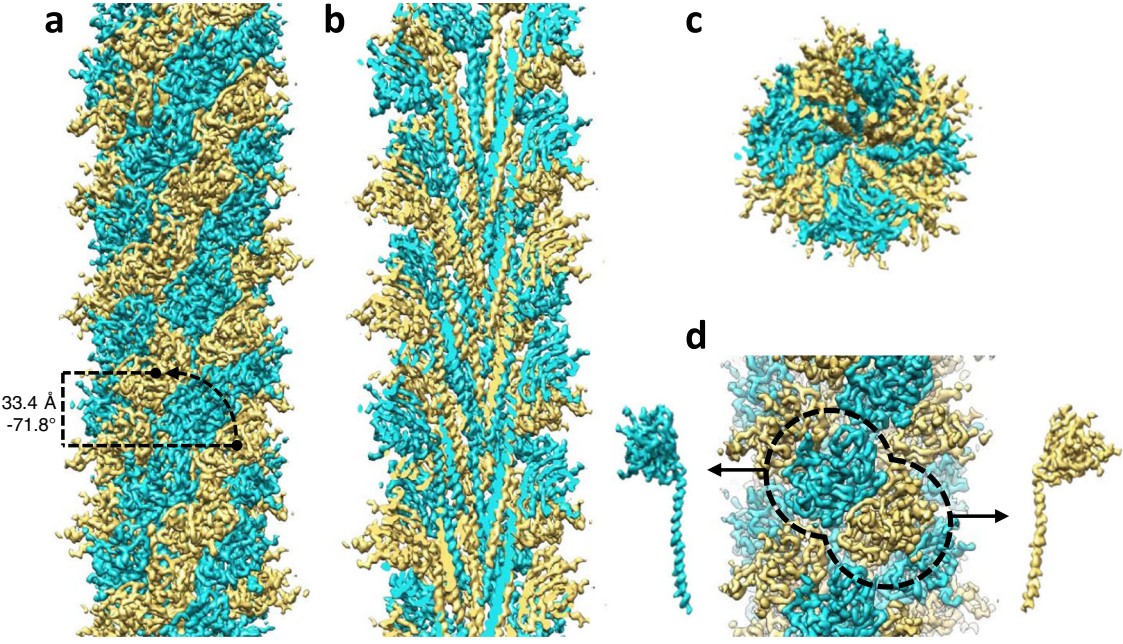

**Fig. 1 CryoEM map of the *M. villosus* archaellum. a** Surface view; **b**, **c** Cross-sections parallel and perpendicular to the filament's long axis, respectively. The two alternating archaellin densities are coloured in cyan (ArlB1) and sand (ArlB2). In **a**, the helical parameters (rise, 33.4 Å; twist, −71.8°) are shown. In **d**, the head domains of the two archaellins are highlighted with a dashed line and the density maps of each subunit are shown left (ArlB1, cyan) and right (ArlB2, sand) of the filament (arrows). Scale bar in **a**, **b** and **c**, 50 Å.

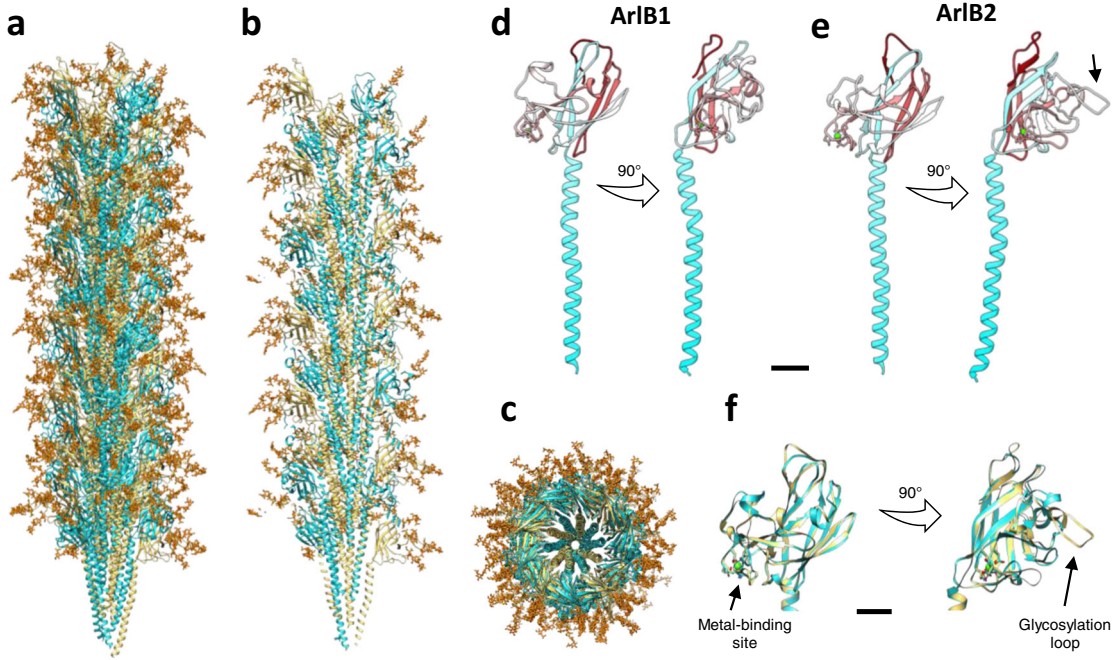

**Fig. 2 Atomic model of the *M. villosus* archaellum filament. a–c** Atomic model of the archaellum filament with glycosylation in side view (**a**) and cross-section parallel (**b**) and perpendicular (**c**) to the filament's long axis. ArlB1 is in cyan, ArlB2 in sand, and glycosylation in orange. Scale bar, 50 Å. **d**, **e** Atomic models of ArlB1 (**d**) ArlB2 (**e**) in ribbon representation and cyan-white-maroon colour scheme (N-terminus, cyan; C-terminus, maroon). The arrow in **e** highlights the "glycosylation loop". **f** Superposition of the ArlB1 (cyan) and ArlB2 (sand) head domain with the metal-binding site and "glycosylation loop" highlighted. Scale bar in **d**, **e** and **f**, 10 Å.

glycosylation sites are located on loops or sites of alternating secondary structure motifs and are equally distributed along the filament's length.

While the head domain of ArlB1 harbours three glycosylation sites ($N_{112}$, $N_{133}$ and $N_{176}$) (Fig. 3b), ArlB2 features six ($N_{70}$, $N_{113}$, $N_{118}$, $N_{133}$, $N_{134}$ and $N_{180}$) (Fig. 3c). Interestingly, two of these glycosylated asparagine residues ($N_{133}$ and $N_{134}$) are consecutive and located in the NNTT motif of the ArlB2 glycosylation loop ($K_{131}$–$A_{137}$).

Since no information on sugar connectivity and identity is available for N-glycans of *M. villosus*, we modelled the branched

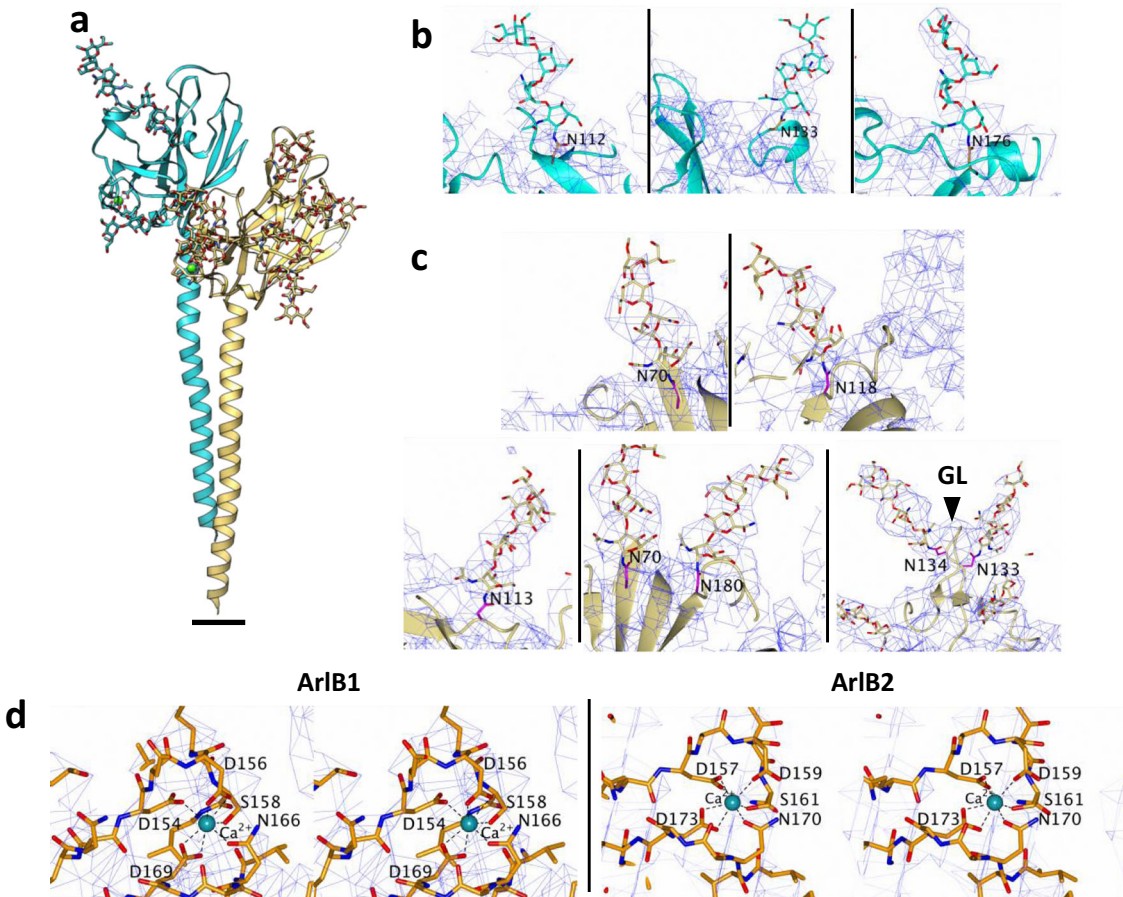

**Fig. 3 Glycosylation and metal-binding sites in ArlB1 and ArlB2. a** Atomic models of ArlB1 (cyan) and ArlB2 (sand) are shown in ribbon representation with modelled glycan molecules, side chains of metal and glycan binding residues shown as stick models and metal ions highlighted as green spheres. Scale bar in **a**, 10 Å. **b** Close-ups of the three glycosylation sites in ArlB1. **c** Close-ups of the six glycosylation sites in ArlB2. GL glycosylation loop. **d** Stereo views of ArlB1 and ArlB2 metal-binding sites. The cryoEM map is represented as a blue mesh and contoured at 2.8 σ. Protein chain is shown as a stick model, Ca$^{2+}$ ion is shown as a cyan sphere with metal-coordinating residues labelled.

heptameric glycan from the related archaeal species *M. thermolithotrophicus*[30] in order to aid model building and refinement. For each of the glycan densities, we were able to fit only the first four sugars of the *M. thermolithotrophicus* polysaccharide. No additional saccharides were resolved beyond the fourth unit, which is likely due to their flexibility or a shorter glycan in *M. villosus*. The fitted glycans provide a detailed model of their location and distribution on the surface of the archaellum (Fig. 2a–c; Fig. 3a–c; Supplementary Fig. 11).

As featured by other archaellins[34], ArlB1 and ArlB2 also contain a metal-binding site located at the periphery of the head domain, in a groove between β6 and β7 in ArlB1 and β7 and β8 in ArlB2 (Fig. 3d). In both archaellins, the metal ion is coordinated by five conserved residues, which differ slightly in their position along the backbone (D$_{154}$, D$_{156}$, S$_{158}$, N$_{166}$, D$_{169}$ in ArlB1 and D$_{157}$, D$_{159}$, S$_{160}$, N$_{170}$, D$_{173}$ in ArlB2). The metal was modelled as a calcium ion. However, due to the prevalence of magnesium over calcium in our growth culture, a mixed population of calcium and magnesium ions cannot be excluded. The metal-binding residues are highly conserved between ArlB1, ArlB2 and the *M. jannaschii* archaellin (Supplementary Fig. 12), for which the site was resolved by X-ray crystallography[34].

**Heteropolymeric assembly of ArlB1 and ArlB2 introduces a screw asymmetry in the archaellum.** To understand the assembly of ArlB1 and ArlB2 in the archaellum structure, we analysed

their organisation and contacts within the filament. Consistent with published structures of homopolymeric archaella[17,20,33] we find that ArlB1 and ArlB2 undergo hydrophobic interactions via their N-terminal tails in the filament's core. In particular, each tail interacts with neighbours at positions n + 3, n − 3, n + 7, n − 7, n − 1, n + 1, n − 4 and n + 4 (Supplementary Fig. 13).

In the outer sheath of the archaellum the head domains of ArlB1 and ArlB2 interact via hydrogen bonds and salt bridges. These occur between alternating ArlB1 and ArlB2 subunits along left-handed 3-start helical strands (Fig. 4a–d; Supplementary Table 2) and along right-handed 7-start strands (Supplementary Fig. 14). The alternating sequence in 3-start direction appears to be favoured by high shape and charge complementarity between ArlB1-2 and ArlB2-1 heterodimers. 13 hydrogen bonds and 3 salt bridges with a ΔG$_{diss}$ of 20.4 kcal/mol are formed at the ArlB1-2 contact zone and 8 hydrogen bonds and 2 salt bridges with a ΔG$_{diss}$ of 16.2 kcal/mol generate the ArlB2-1 interface (Fig. 4c; Supplementary Table 2). In contrast, poor charge and shape complementarity would not allow the formation of ArlB1-1 and ArlB2-2 in the 3-start direction. Interestingly, the sequence of alternating subunits is not always in sync in adjacent 3-start helices. One of the helical strands is shifted by one subunit, moving it out of the register (Fig. 4b). This results in an anisotropic distribution of ArlB1 and ArlB2 throughout the archaellum, meaning that ArlB1 and ArlB2 neither alternate nor form perfect homopolymeric strands in any direction but 3-start.

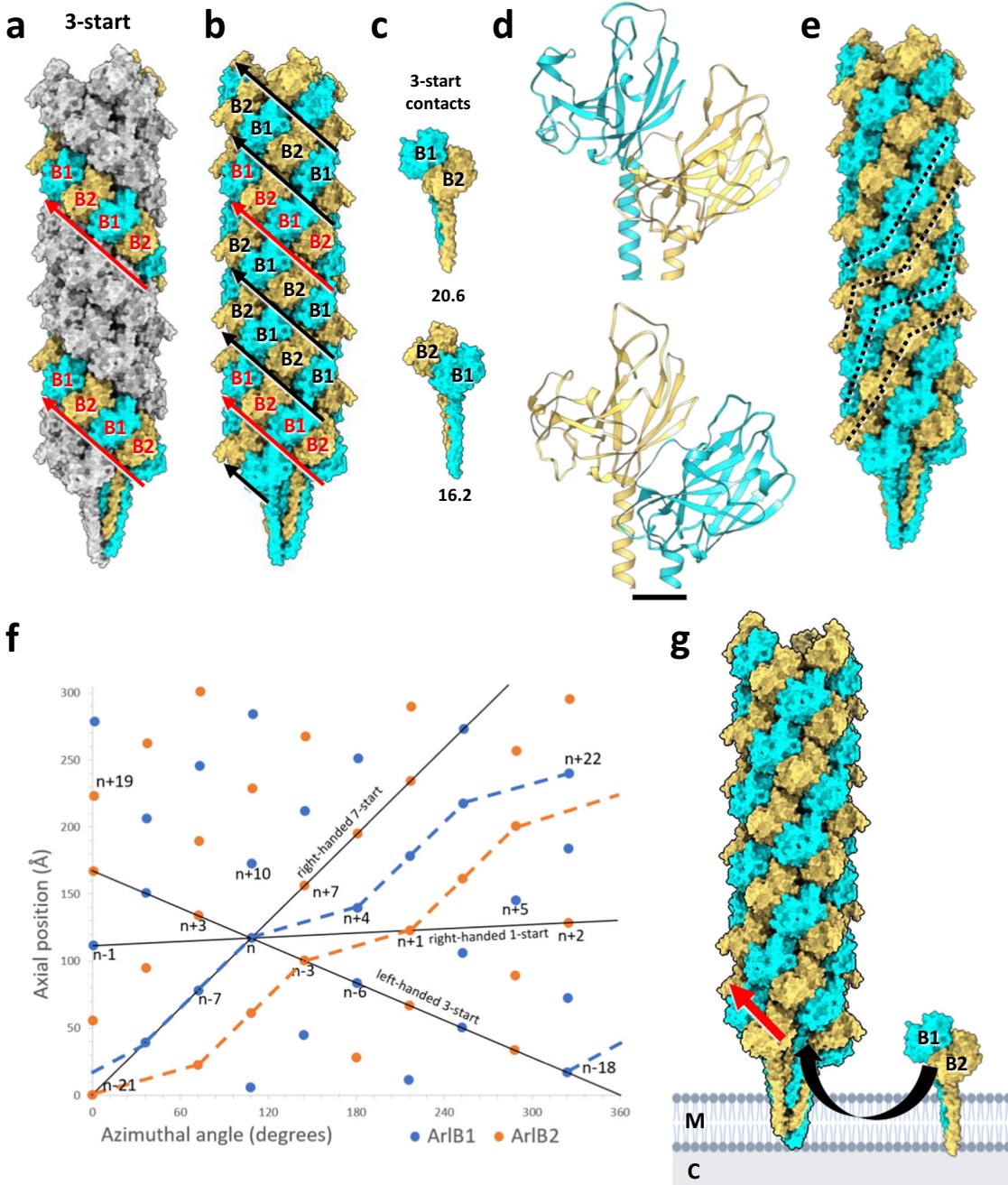

**Fig. 4 Screw axis asymmetry in the *M. villosus* archaellum. a** The archaellum atomic model viewed parallel to the filament's axis. ArlB1 (cyan) and ArlB2 (sand) subunits are highlighted along one left-handed 3-start helical strand. ArlB1 and ArlB2 alternate in the 3-start direction (indicated by the red arrows). **b** Subunits within all three 3-start strands are highlighted. Every third 3-start strand (red arrow) is out of register with respect to the other two 3-start strands (black arrows). **c** Intermolecular contacts in 3-start direction between ArlB1-2 (top) and ArlB2-1 (bottom) dimers. The dissociation energies ($\Delta G_{diss}$ calculated by PISA) for both interactions are indicated below each dimer (unit: kcal/mol). **d** Close-ups of the atomic models (head domains) of ArlB1-2 and ArlB2-1 as in **c**. **e** Tracing subunits of the same kind through the filament indicates non-helical homopolymeric pseudo-strands (dashed lines). **f** Helical net diagram showing the positions of ArlB1 (blue dots) and ArlB2 (orange dots) in a two-dimensional plot. Solid black lines show various component helices. Blue and orange lines trace non-helical homopolymeric pseudo-strands of ArlB1 and ArlB2. **g** Model of the hypothetical assembly of the heteropolymeric archaellum from pre-formed ArlB1-2 dimers that are added to the growing filament in 3-start direction. M, membrane; C, cytosol; red arrow, 3-start direction. For simplicity, the archaellum machinery and the S-layer have not been drawn. Scale bar in **d**, 10 Å.

Along the left-handed 7-start helical strands the two subunits undergo all four possible contacts (ArlB1-2, ArlB2-1, ArlB1-1, ArlB2-2) (Supplementary Fig. 14). The interactions in the right-handed 7-start direction are by an order of magnitude weaker than those in the 3-start direction ($\Delta G_{diss}$ of 1.7–4.7 kcal/mol), as they are stabilised by only 2–5 salt bridges per interunit interface

and limited hydrophobic contact between the tails (Supplementary Table 2).

Strikingly, when tracing subunits of the same type (ArlB1 or ArlB2 only) throughout the filament, the archaellum appears to be defined by homopolymeric right-handed pseudo-strands with broken helical symmetry (Fig. 4e, f). Here, the symmetry of the

pseudo-strands is disrupted at the interface between every third and fourth subunit, giving the pseudo-strands a "stepped" appearance, in contrast to a hypothetical isotropic filament (with n + 2 geometry) (Supplementary Fig. 5c, d, g).

**The flexibility of the archaellum filament**. The archaellum rotates, prompted by an ATP-driven intracellular motor that generates torque. To propagate the motion along its length, the filament transitions from a curved filament in resting cells[20] (Supplementary Fig. 15) into a rotating superhelix when cells swim[41]. However, due to the nature of helical image processing of cryoEM data, all available structures of archaella suggest idealised helical filaments that are infinitely straight.

In order to obtain a truer representation of the structure of the archaellum and to probe the molecular basis for its flexibility, we performed 3D variability analysis in cryoSPARC, which estimates molecular motion from micrographs[42]. This analysis resulted in a series of 20 maps, displaying a distinctly bent conformation of the archaellum (see Supplementary Movie 3 for morphing between these conformations). Refining our atomic model based on individual conformations, we visualised the dynamic properties of the archaellum at the molecular level (Fig. 5; Supplementary Movies 4–7).

Our analysis reveals that as the filament bends, the subunits are compressed at its concave side and stretched apart along its convex side (Supplementary Movies 3, 4). To allow for this flexibility, the tails in the backbone rotate around and slide along the filament's axis (Fig. 5a, b; Supplementary Movie 5), whereas the head domains swing diagonally to the filament's axis (Fig. 5h; Supplementary Fig. 16). Along pseudo-helical ArlB1 and ArlB2 strands (Fig. 5c), the head domains of both archaellins are flexible, and their trajectories differ depending on the subunit location within the filament (Fig. 5d–g). Each head domain is free to swing in all directions within the cylinder surface of the outer sheath and with respect to the tail domain. An example of the head trajectories of two homopolymeric strands is shown in Fig. 5e, g and Supplementary Movies 6, 7. Notably, we do not see substantial flexibility or conformational changes within the head or the tail domains themselves, indicating that they move relative to each other as rigid bodies. Moreover, no significant difference in flexibility between ArlB1 and ArlB2 was observed.

The flexibility of each archaellin is enabled by a hinge of two residues ($Ser_{60}$, $Gly_{61}$) linking the head and tail of each subunit. This hinge is highly conserved in ArlB1 and ArlB2 sequences within the archaeal kingdom (Fig. 5i; Supplementary Fig. 17a). Interestingly, the sequence of this region is partially conserved, albeit different from archaella, in archaeal T4P (Supplementary Fig. 17b, d). In bacterial T4P a conserved loop region coincides with the melted central portion of the N-terminal α-helix in the core of the pilus (Supplementary Fig. 17c, d).

## Discussion

Many archaea encode for multiple archaellins, which likely originated from gene duplication events[43]. Studies employing genetics, biochemistry and molecular biology have shown that while some archaella contain a single major archaellin, others are assembled from two homologues[26,29–32]. The three structures of archaellar filaments that have been published so far suggest homopolymers[20,33,34], and until now it was unknown how two or more distinct proteins can assemble into a single archaellum.

Here, we present the structure of a heteropolymeric archaellum. The filament contains two archaellin homologues (ArlB1 and ArlB2), which became distinguishable when the cryoEM map was reconstructed without imposing helical symmetry. This finding highlights the importance of relaxing the symmetry

during the processing of any helical data, as structural heterogeneities may otherwise be overlooked. In this regard, it would be interesting to revisit previously published structures of archaella and use our image processing strategy to investigate if these are indeed homopolymers by nature or perhaps also consist of two alternating subunits.

The evolutionary advantages of a heteropolymeric over homopolymeric archaella are not fully understood. However, accumulating evidence suggests that these include increased cellular motility and filament stability. For example, while individual ΔarlB1 and ΔarlB2 knockout mutants in *H. lacusprofundi* DL18 still assemble archaella, they are less salt-resistant than wild-type[32]. In addition, micrographs of negatively stained filaments suggested that ΔarlB1 archaella are more flexible and ΔarlB2 archaella are stiffer compared to wild-type. This is in line with the different binding energies that we have calculated for the *M. villosus* archaellum, where ArlB2-2 interfaces in 7-start direction are significantly more stable than the corresponding ArlB1-1 contact (Supplementary Table 2). Provided that *M. villosus* could form homopolymeric archaella, an ArlB2-only filament would thus likely be more stable but stiffer than an ArlB1 homopolymer. Notably, knocking out one of the two archaellar subunits in *H. salinarum* results in reduced motility[31] and ablating either of the two archaellins in *M. maripaludis* abolishes archaella and swimming motility entirely[29]. Furthermore, in *Haloarcula marismortui* the two archaellins ArlA2 and ArlB are ecoparalogues, meaning that different archaella are assembled in response to varying environmental situations (in this case the level of salinity in the environment)[44]. This suggests that the ability to form heteropolymeric archaella can, at least in some species, add to the adaptability of the cell to varying habitat conditions.

In the heteropolymeric archaellum of *M. villosus*, ArB1 and ArlB2 are not isotopically organised. We propose that the resulting screw axis asymmetry, together with the helical variation of rigidity along the 3-start direction, enhances the supercoiling propensity of the filament. This would explain how the archaellum forms the observed stable right-handed supercoil independent of the direction of gyration[41], as a prerequisite for cellular propulsion in opposite directions.

The published structures of T4P in both archaea and bacteria report homopolymeric filaments constituted by one major pilin[45–49]. In one example, however, two different homopolymeric T4P filaments are assembled simultaneously and by the same machinery[49]. Only one study published to date provides evidence of a heteropolymeric T4P in a Gram-positive bacterium, but the arrangement of the two subunits within the filament is speculative[50]. These findings suggest that the specialised structures of pilins/archaellins may allow for unique patterns of assembly within filaments, and differences within the T4F superfamily may be more widespread than originally thought.

Archaella are assembled by the platform protein ArlJ, powered by the ATPase ArlI[6]. For homopolymeric archaella, the filament is thought to be built from monomers. However, building the complex n + 6 symmetry observed in the *M. villosus* archaellum from archaellin monomers would need to be selective for alternating subunits, and thus require a more complex assembly mechanism compared to one that assembles homopolymeric archaella. Therefore, it is conceivable that the filament is instead assembled by pre-formed ArlB1-2 heterodimers, which are then added to the 3-start strands of the growing archaellum (Fig. 4g). This is in line with our observation that ArlB1-2 interfaces show by far the strongest interactions across the filament, which suggests that ArlB1-2 may be the fundamental building block of the *M. villosus* archaellum. Notwithstanding this, knockout mutants of one of two major archaellins still assemble archaella[32]. This

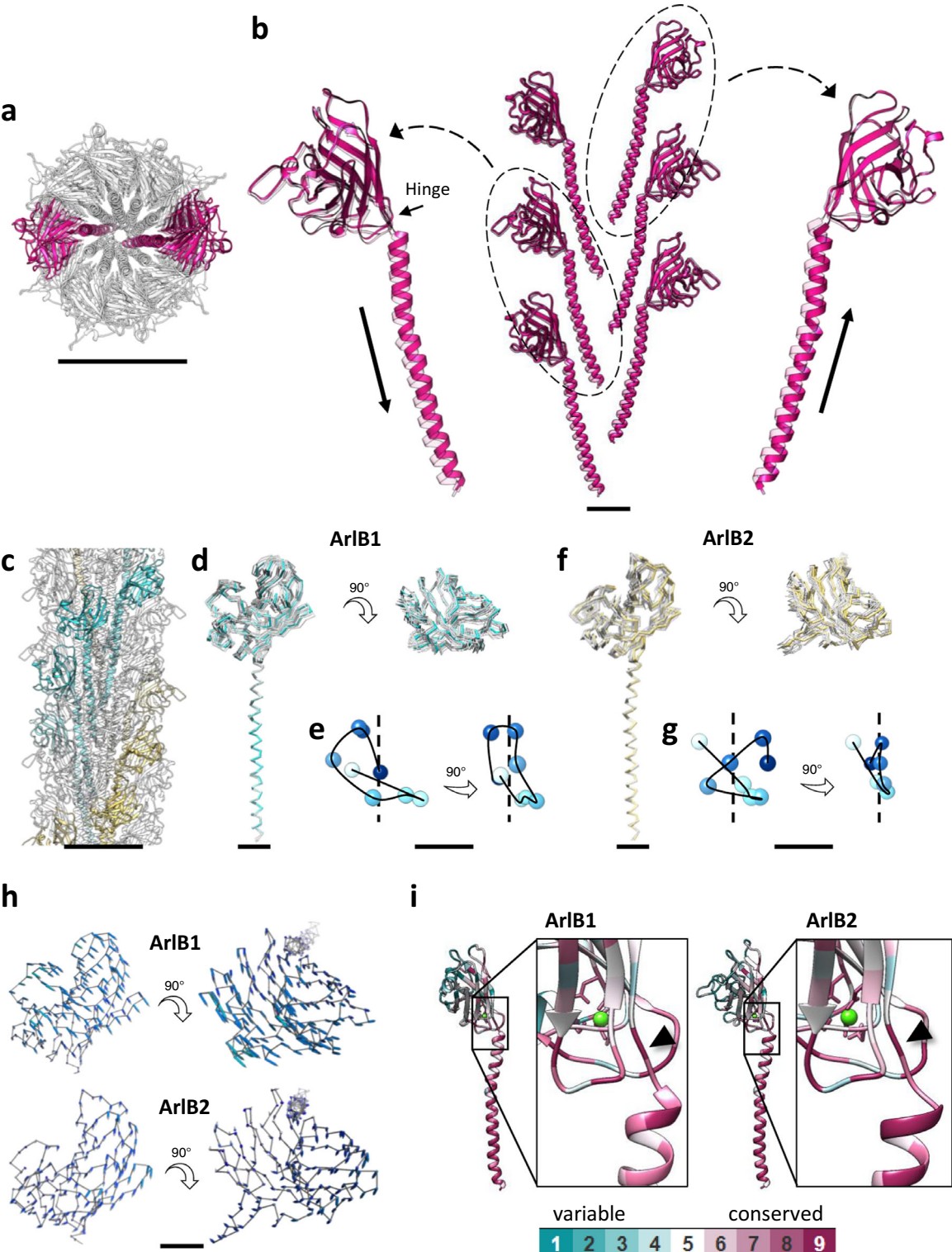

**Fig. 5 Flexibility of the *M. villosus* archaellum. a**, **b** Atomic model showing the motion of the tail domains. Two opposite n + 10 sets of protein monomers (**a**) were selected and the head domains of the constituting monomers were aligned. The solid and transparent magenta models (**b**) were fitted into the frame 0 and frame 19 maps of the cryoSPARC 3DVA, respectively. The arrows along the tail domains in **b** highlight the up-down motion of the tail domains of opposite n + 10 strands. **c**–**g** Atomic models and motion trajectories showing the flexibility of the head domains along a full filament turn. Eight ArlB1 and eight ArlB2 monomers forming a full helical turn were selected (**c**) and their tail domains aligned. Panels **d**, **f** show the flexibility of the head domains of ArlB1 and ArlB2, respectively. Panels **e**, **g** represent the motion trajectories of the centroids (shades of blue) of each head domain. The black curve highlights the trajectory, and the dotted black line represents the filament axis. **h** Atomic models of one ArlB1 and one ArlB2 head domains showing their displacement during filament motion. The motion is visualised by vectors linking the backbone atoms of the atomic models fitting into the frame 0 (grey line) and frame 19 (not shown for better clarity) maps of the cryoSPARC 3DVA. **i** Estimation of the evolutionary conservation of AlrB1 and ArlB2. Insets focus on the hinge region. Scale bars in **a** and **c**, 50 Å; in **b**, **d**, **f**, **h**, 10 Å; in **e** and **g**, 2 Å.

shows that the assembly platform ArlJ is also capable of assembling monomers of the same kind into homolopolymeric archaella, potentially adding to the functional versatility of the archaellum machinery.

While the function of minor pilins in bacteria has been more extensively investigated, the role of minor archaellins remains elusive. In bacteria, minor pilins have been shown to initiate pilus assembly, by forming a cap at the tip of the pilus, as well as promote pilus functions, such as DNA binding, aggregation, and adherence[51]. For some archaeal species, including *M. maripaludis*[29], *M. voltae*[35], *M. thermolithotrophicus*[30], *H. salinarum*[36] and *M. hungatei*[52] (Supplementary Fig. 18), it has been shown that the archaellum filament is linked to the basal body through a distinct structure described as reminiscent of the flagellar hook found in bacteria.

It is conceivable that the remaining minor archaellin, ArlB3, assembles into a terminal structure of the archaellum, which could either act as a cap or basal structure. Corroborating this notion, sequence alignments show a significant degree of conservation between *M. villosus* ArlB3 and the ArlB3 homologs from *M. voltae* (48.3% identity and 78.5% similarity) and *M. maripaludis* (50.4% identity and 75.4% similarity). In contrast, a lesser degree of conservation is observed between *M. villosus* ArlB1/ArlB2 with ArlB3 from *M. voltae and M. maripaludis* (Supplementary Fig. 19).

One of the most obvious differences between the two archaellin components of the heteropolymeric *M. villosus* archaellum is the number of glycosylation sites. Interestingly, ArlB2 shows glycosylation sites on two consecutive Asn residues within the NNTT motif in its glycosylation loop. The sharply bent structure of the loop minimises the steric hindrance between the adjacent glycans. The NNTT motif is common in viral proteins[53] but simultaneous glycosylation of two neighbouring residues has so far been shown only for the Gp3 protein of the Equine Arteritis Virus, in which glycosylation may prevent processing of an adjacent cleavage site[54].

Glycosylation plays a key role in archaellum filament assembly and motility[55–59]. In addition, it has been suggested that glycosylation increases the resilience of archaeal T4F against acidic habitats[45,46], as well as thermal stability, with the latter being directly proportional to the degree of glycosylation[60]. *M. villosus* thrives at an optimal temperature of 80 °C. The glycosylation that we observe in the *M. villosus* archaellum could be an adaption to hot environments. Moreover, because *M. villosus* can grow between 55 and 90 °C, an interesting hypothesis to test would be to observe if at different temperatures archaella can be assembled by one or the other archaellin homologues independently. If this were the case, assembling archaella with different homologues could be a way to modulate the glycosylation level of the archaellum filament in response to different temperatures.

Archaella and bacterial flagella propel the cell in a process that starts from the motor complex, where torque is generated, and is transmitted through into the filament[6]. The bacterial flagellum filament supercoils due to its constituting subunits switching between two discrete states, long and short[61]. In archaella the conformational changes of the filament subunits are widely unexplored.

Here we provide direct insights into the molecular flexibility of the archaellum at near-atomic detail and identify two major structural elements that govern the filament's flexibility. At the monomer level, the loop region between the head and the tail domain acts as a hinge, allowing the two domains to move relative to each other. The amino acid sequence of the hinge region is highly conserved in archaellins and differs from that of archaeal T4P. This indicates that the hinge region of archaellins is specialised to filaments that perform a propulsive motion. The amino acids that make up the loop of non-motile archaeal pili

contain larger side chains (QQT/QVT/QGT) than those of archaella (SG). This may infer more steric hindrance and less flexibility of pili-borne compared to archaellar hinges. Based on this, the archaeal hinge region could be used to distinguish between archaellins and archaeal type-4 pilins when predicting their function at the sequence level. In bacterial type-4 pilins, the globular head is packed against the N-terminal α-helix, likely resulting in further decreased flexibility of this area compared to archaellins or archaeal pilins. Instead, bacterial type-4 pilins contain a conserved "melted" region within the N-terminal α-helix, which could aid their flexibility in a similar fashion[48].

At the core of the archaellum, the tail domains of individual subunits slide along and rotate around each other, further increasing the flexibility of the archaellum. This sliding motion is enabled by the hydrophobic interactions that solely hold together the tail domains. This type of intermolecular interaction is known to provide "molecular grease" in various flexible or rotary protein assemblies[62]. Together, the hinges within each subunit, as well as the molecular grease within the filament's core provide the basis for the flexibility of archaella that is essential in their function as molecular propellers. It is important to point out that the flexibility we observe is subtle, and that further biological validation is needed to undoubtfully link it to the functional filament motion.

The flexibility of the *M. villosus* archaellum filament supports recent insights into the motion of the flagellar hook from the bacterium *S. enterica*[63,64]. Similar to archaellins, the three domains of the flagellar hook subunits (D1, D2, D3) behave as rigid bodies connected by two flexible hinges. The authors identified eleven distinctive conformations of the subunits along with a full helical turn. In contrast, our data suggest that archaellins change their shape in a continuous manner. While in the flagellar hook discrete conformations of the subunits may be important to maintain its hooked shape, the continuity of motion seen in archaellins is likely a prerequisite for continuously undulating and supercoiling archaella.

## Methods

**Cell culture**. Anaerobic media were prepared according to the technique described by Balch and Wolfe[65]. Dissolved media components were flushed with $N_2/CO_2$, reduced with $Na_2S$, and pH adjusted. Twenty millilitres were dispensed to 120 ml-serum bottles in a vinyl anaerobic chamber (Coy Laboratory Products, Inc., Grass Lake, MI, USA). Rubber stopper-sealed and crimped bottles were pressurised with 3 bar (absolute pressure) $H_2/CO_2$ (80:20, v/v) as gas phase using a gassing manifold (three vacuum and pressurisation cycles). *M. villosus* was originally isolated from hot sediments sampled during the RV Poseidon cruise in 1997 to the Kolbeinsey Ridge[66]. Standard incubation was done in serum bottles overnight at 80 °C[11]. For archaella isolation, cells were grown in a 50 l bioreactor (Bioengeneering, Wald, Switzerland) using MMC medium designed for the fermentation process. The bioreactor was inoculated with 2.5 ml of *M. villosus* cells and supplied continuously with 2.5 bar (absolute pressure) $H_2/CO_2$ (80:20, v/v) as the gas phase. The temperature was kept constant at 80 °C and the pH at 6.5. Cell densities of grown cultures were determined by direct cell counting using a Thoma counting chamber (depth 0.02 mm). Standard reagents were purchased from VWR or Merck. The medium consisted of (per l in distilled water): NaCl (430.0 mM), $NaHCO_3$ (10.0 mM), $MgCl_2 \times 6H_2O$ (38.0 mM), KCl (3.6 mM), $NH_4Cl$ (22.1 mM), $CaCl_2 \times 2H_2O$ (2.5 mM), $K_2HPO_4$ (0.3 mM), $KH_2PO_4$ (4.1 mM), $(NH_4)2Fe(SO_4)_2 \times 6H_2O$ (30.6 μM), $(NH_4)2Ni(SO_4)_2 \times 6H_2O$ (43.5 μM), $Na_2MoO_4 \times 2H_2O$ (10.1 μm), $Na_2WO_4 \times 2H_2O$ (9.9 μM), $Na_2SeO_4$ (42.3 μM), 10× tracemineral solution (1.00 ml), $Na_2S \times 7\text{-}9H_2O$ (0.40 g). The pH was adjusted to 6.5 with 50 ml $H_2SO_4$ (diluted 1 + 1 in MilliQ). The tracemineral solution 10-fold[67] contained per litre: $MgSO_4 \times 7H_2O$ (121.7 mM), NaCl (171.1 mM), $MnSO_4 \times H_2O$ (29.6 mM), $(NH_4)_2Ni(SO_4)_2 \times 6H_2O$ (7.1 mM), $CoSO_4 \times 7H_2O$ (6.4 mM), $ZnSO_4 \times 7H_2O$ (6.3 mM), $FeSO_4 \times 7H_2O$ (3.6 mM), $CaCl_2 \times 2H_2O$ (6.8 mM), $AlK (SO_4)_2 \times 12H_2O$ (0.4 mM), $CuSO_4 \times 5H_2O$ (0.4 mM), $H_3BO_3$ (1.6 mM), $Na_2MoO_4 \times 2H_2O$ (0.4 mM), $Na_2WO_4 \times 2H_2O$ (0.3 mM), $Na_2SeO_4$ (0.5 mM). Trace element solution was titrated at pH 1.0 with 1 M $H_2SO_4$ before dissolving chemicals, and it was filter sterilised and stored in the dark at 4 °C.

**Preparation of archaella**. The archaella preparation protocol was adapted from Kalmokoff et al.[28] and Näther et al.[10]. The complete bioreactor was used, and cells

were harvested at the early stationary phase (7000–8000 × *g*) and concentrated (3500 × *g*, 30 min, 4 °C; Sorvall RC 5 C plus, rotor GS3). A total of 17.5 g of cells (wet weight) were obtained. Cells were resuspended in a small amount of medium and then archaella were detached by shearing force and cavitation, causing disruption of the cells (ULTRA-TURRAX T25 high-speed homogeniser; 1 min at 13,000 rpm and 10 sec at 22,000 rpm). Cell debris was separated from archaella by centrifugation (34,500 × *g*, 4 °C; Sorvall RC 5 C plus, rotor SS34) for 20 min. Archaella were pelleted from the supernatant by ultracentrifugation (60,000 × *g*, 90 min, 4 °C; Beckman Optima LE-80K, rotor 70Ti), resuspended in 150 µl 0.1 M HEPES/NAOH pH 7.0 and purified for 48 h (250,000 × *g*, 4 °C; Beckman Optima LE-80K, rotor SW60-Ti) using a CsCl gradient (0.45 g/ml; Biomol). Fractions were taken by puncturing the ultracentrifuge tubes with sterile syringes, dialyzed against aerobic ½ SME/5 mM HEPES/NaOH pH 7.0, and analysed by SDS-PAGE and transmission electron microscopy[11] to identify the archaella-containing band. After addition of sodium azide to a final concentration of 0.001% (w/v), samples were stored at 4 °C.

**CryoEM of archaella and helical processing.** Prior to freezing, a 3 µl drop of undiluted suspension containing purified archaella was applied to glow-discharged 300 mesh copper R2/2 Quantifoil grids. The grids were blotted with 597 Whatman filter paper for 4 sec, using −1 blot force, in 95% relative humidity, at 21 °C and plunge-frozen in liquid ethane using a Mark IV Vitrobot (FEI). Dose-fractionated movies were collected in linear mode using a 300 kV Titan Krios equipped with a Falcon III detector (Thermo Fisher Scientific) (Supplementary Table 1). 39-fraction movies were recorded using EPU 1.5 (Thermo Fisher Scientific) with an exposure time of 1 sec and a total electron dose of 37 e⁻/Å² at a defocus range of −2.3 to −1.1 µm (0.3 µm increment). A total of 2759 movies were collected and processed using the Relion 3.1 pipeline[38] and finalised using cryoSPARC 3.1.0[39]. Briefly, the movies were motion-corrected using MotionCorr 2[68] and CTF estimation was carried out using CTFFind 4[69]. Helical segments were manually picked using Helixboxer from the EMAN-2 package[70]. The coordinates of the picked particles were imported into Relion 3.1, where particle extraction, 2D classification, 3D classification, 3D refinement, B-factor sharpening, CTF-refinement and Bayesian polishing were performed. Each filament was subdivided into 256 × 256 pixel size boxes at an offset of 6 Å, resulting in 929,165 segments upon particle extraction. The dataset was then 3D classified (8 classes) and the best class (containing 399,178 segments) was selected for 3D refinement using the helical parameters published for the *P. furiosus* archaellum[20]. The refined consensus map was improved using CTF-refinement and Bayesian polishing in Relion 3.1. The shiny particles obtained from the Bayesian polishing were further refined using the optimised helical parameters 5.57 Å rise and 108° twist after helical refinement. The resulting map had a resolution of 3.29 Å. The particles of the final Relion 3.1 helical refinement were imported into cryoSPARC 3.1.0 and a new refinement was performed using the helical refinement (BETA) algorithm without imposing helical parameters. This resulted in a reconstruction at a resolution of 3.28 Å by gold-standard FSC in which two distinct subunits could be distinguished. To corroborate these results, another round of 3D refinement with no helical symmetry was performed in Relion 3.1, which provided a comparable result. The two subunits alternate along the left-handed 3-start helices as shown on the archaellum net diagram (Fig. 4f, Supplementary Fig. 5a, e). An attempt to perform a helical refinement with (n + 2) helical parameter 11.14 Å rise and 216 ° twist resulted in a filament reconstruction with lower resolution and poorly resolved features. Further inspection of the net diagram allowed to establish the minimal transformation of the *M. villosus* archaellum in which each monomer superimposes onto its equivalent, and which proved to be n + 6. A final helical refinement was performed in cryoSPARC 3.1.0, this time imposing a (n + 6) helical parameters 33.4 Å rise and −71.8° twist (see results below). This produced a final map of 3.08 Å resolution by gold-standard FSC 0.143. DeepEMhancer was used for final map sharpening[71].

**Model building and validation.** Initially, the *P. furiosus* ArlB0 archaellin structure (PDB ID: 5O4U) was positioned into the Relion 3.1 cryoEM map, refined using REFMAC5[72] and rebuilt to match the *M. villosus* ArlB2 sequence using *Coot*[73]. Both candidate sequences (ArlB1 and ArlB3) were attempted to be fitted by Buccaneer[74] into the density of the second archaellin found in the higher resolution cryoSPARC 3.1.0 cryoEM map. The sequence match the cryoEM map features and glycosylation pattern unambiguously identified ArlB1 as the second protein forming the archaellum. Both ArlB1 and ArlB2 atomic models were then validated using Isolde[75]. Two full turns of the archaellum filament were built in UCSF Chimera[76] using ArlB1-ArlB2 3-start dimers. Optimised helical parameters with the position of each monomer were rigid-body refined in *Coot* to adjust for local variations in the non-helically refined map. A script containing a number of CCP4[77] programs was prepared to propagate any modification to a structure of a single monomer into a full filament. The resulting model was refined using REFMAC5 in CCP-EM interface[78]. Glycans were modelled in *Coot* using the sugar unit identity and connectivity reported for the glycan structure of *M. thermolithotrophicus*. This structure was recently reported as α-d-glycero-d-manno-Hep3OMe6OMe-(1–3)-[α-GalNAcA3OMe-(1–2)-]-β-Man-(1–4)-[β-GalA3O-Me4OAc6CMe-(1–4)-α-GalA-(1–2)-]-α-GalAN-(1–3)-β-GalNAc-Asn[30]. Restraints dictionaries for non-standard sugars were prepared using JLIGAND[79].

The calcium ion in ArlB1 and ArlB2 was modelled on the basis of the metal coordination and distances observed in our structure and agreed with the metal assignment in[34].

**3D variability analysis and molecular flexibility.** 3D variability analysis (3DVA)[42] was performed in cryoSPARC 3.1.0 using the 399,178 helical segments from the cryoSPARC 3.1.0 helical refinement. First, the symmetry expansion step was run. Then, the particles stack created by the symmetry expansion was used for 3DVA. Different lowpass filter values were tested (3.5, 5, 8 and 10 Å) in simple mode with 20 intermediate frames and with 10 variability components as output. All components showed the same type of motion.

In order to investigate the filament's flexibility at the molecular level, we selected a 3DVA component obtained at 3.5 Å lowpass filter and used the first frame (number 0) and last frame (number 19) of the component corresponding to two cryoEM maps at the two extremes of the filament motion range. The filament's atomic model was positioned into the two cryoEM maps using phased molecular replacement in MOLREP[80] and refined using Phenix[81], thereby producing two atomic models fitting the two cryoEM maps. The two atomic models were finally morphed using the morphing function in UCSF Chimera to display the full conformational space of the filament (Supplementary Movie 4), in particular the tail domains (Fig. 5a, b; Supplementary Movie 5). To show the local flexibility of the head domains, one ArlB1 and one ArlB2 subunits were isolated from the two atomic models fitting the two cryoEM maps at frame 0 and frame 19 and r.m.s.d-coloured vectors were calculated linking the Cα of the head domains (Fig. 5h). To investigate the flexibility of the head domains along the filament, two homopolymeric strands of ArlB1 and ArlB2 were isolated from the atomic model fitting the cryoEM map at frame 0. Then for each strand the tails of its constituting subunits forming a full filament turn were aligned (Fig. 5c–g; Supplementary Movie 6, 7).

**STEM tomography.** Cells were cryo-immobilised using high-pressure freezing (Leica HPM 100, Munich), then freeze-substituted in ethanol containing 0.5% glutardialdehyde, 1% formaldehyde, 0.5% uranyl acetate, 5% water and finally embedded into Spurr resin. Semithin sections with a thickness of 800–900 nm were cut using a diamond knife (Diatome) on an ultramicrotome (Leica UC6). Sections were mounted on formvar-coated grids (100 mesh Cu grids; Plano, Wetzlar). For STEM tomography at 200 kV[82], a 3 nm carbon layer (Cressington turbo 208 Carbon) was evaporated onto one side of the sections. Fiducials (protein A-gold, 15 nm; University Medical Center, Utrecht, NL) were added onto both sides of the sections, to aid the alignment and reconstruction process. STEM tomography was done on a JEM-2100F (JEOL GmbH, Freising, Germany) using an electron beam with a small semi-convergence angle, essentially as described[82]. Tiltseries (STEM bright field) were recorded from +66 deg to −66 deg, using the EM-TOOLS software (TVIPS, Gauting, Germany). Reconstruction of 3D volumes was done in IMOD (Univ. of Colorado at Boulder, USA) using the weighted back projection, incl. a SIRT-like filter equivalent to 15 iterations.

**Sequence analyses.** The evolutionary conservation of amino acids positions in AlrB1 and ArlB2 (Fig. 5i) and bacterial T4P (Supplementary Fig. 17c) was estimated using the ConSurf server[83]. The sequence alignment and secondary structure assignment of *M. villosus* ArlB1 and ArlB2 were performed with BLASTp and visualised by ESPript3[84] (Supplementary Fig. 9). The archaellar operons (Supplementary Fig. 18) were visualised using ARTEMIS[85] and the automatic genome annotations were downloaded from NCBI. The sequence comparison and alignment between archaellins and archaeal T4P (Supplementary Fig. 17a,b), and *M. maripaludis* ArlB3, *M. voltae* ArlB3 and *M. villosus* ArlB1, 2, 3 archaellins (Supplementary Fig. 19a) were performed with BLASTp and the Praline server[86].

**Structure analysis and presentation.** The structure of the *M. villosus* archaellum filament was visualised using UCSF Chimera, Chimera-X[87], Coot and CCP4mg[88]. The topology diagrams were prepared using Pro-origami[89] and TopDraw[90]. Interfaces between different ArlB1 and ArlB2 monomers in the archaellum were analysed using the PISA software[91]. The Molprobity score in Supplementary Table 1 was calculated using[92].

**Reporting summary.** Further information on research design is available in the Nature Research Reporting Summary linked to this article.

## Data availability

The archaellum atomic coordinates and electron density map generated in this study have been deposited in the Protein Data Bank (https://www.rcsb.org/) with accession number 7OFQ and in the EM DataResource (https://www.emdataresource.org/) with the accession number EMDB–12875. The raw movies have been deposited in the EMPIAR database (https://www.ebi.ac.uk/empiar/) with accession number EMPIAR-10884. Other structural data used in this study are: *P. furiosus* ArlB0 (PDB ID: 5O4U), *M. jannaschii* ArlB1 (5YA6), *T. thermophilus* PilA4 (PDB ID: 6XXD), *P. arsenaticum* PilA (PDB ID: 6W8U).

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

## Acknowledgements

We thank the RMS for providing microscope time as part of the Single Particle Cryo-TEM school, University of Leeds. The FEI Titan Krios microscopes were funded by the University of Leeds (UoL ABSL award) and the Wellcome Trust (108466/Z/15/Z). We also acknowledge Alexander Neuhaus for his invaluable assistance with helical reconstruction; Erik Zupa for providing us with the PyMOL script to plot r.m.s.d.-coloured vectors, and Simone Antonio De Rose for help with helical net diagrams. Molecular graphics and analyses were performed with UCSF Chimera, developed by the Resource for Biocomputing, Visualization,and Informatics at the University of California, San Francisco, with support from NIH P41-GM103311. We are grateful to Yvonne Bilek, Thomas Hader and Konrad Eichinger for their help with mass cultivation of *M. villosus* cells. For this project, L.G., B.D. and M.M. were funded by the European Research Council (ERC) under the European Union's Horizon 2020 research and innovation programme (grant agreement No 803894). R.C. was funded by the University of Exeter start up funds awarded to V.G., and a Wellcome Trust Seed Award in Science (210363/Z/18/Z) awarded to V.G.; A.B. and R.R. received funding from the Deutsche Forschungsgemeinschaft (WI 731/10-1) awarded to R.R. and Reinhard Wirth. The authors would like to dedicate this study to the memory of Reinhard Wirth, who supervised initial studies on *M. villosus* archaella.

## Author contributions

L.G.: Performed the research, provided methodology, wrote the manuscript. M.I.: Performed the research, provided methodology, wrote the manuscript. R.C.: Provided methodology. M.M.: Provided methodology. V.G.: Provided resources and funding for R.C., contributed to writing of the manuscript. A.B.: Provided resources and methodology, contributed to writing of the manuscript. R.R.: Provided resources and methodology. B.D.: Provided funding for L.G., performed research, provided resources, wrote the manuscript.

## Competing interests

The authors declare no competing interests.
