## [Peer Review File · Nature Communications]

An archaellum filament composed of two alternating subunitsREVIEWER COMMENTS

Reviewer #1 (Remarks to the Author):

The manuscript entitled "New insights into the architecture and dynamics of archaella" by Gambelli et al. focuses on the archaellum in *Methanocaldococcus villosus*. Archaella are swimming filaments, functional analogs of bacterial flagella but belonging to the superfamily of type 4 filaments (T4F). In this paper, the authors present the structure of the *M. villosus* archaellum determined by cryoEM at 3.08 Å resolution. This is the first heteropolymeric structure of a T4F since it consists of two alternating subunits, ArlB1 and ArlB2. There is evidence that this heteropolymeric organisation might be quite frequent in the T4F superfamily. They propose a very interesting model in which an ArlB1-ArlB2 heterodimer, that exhibits extensive intermolecular interactions, would be the building block of the filament. The paper is well written and the experiments are carefully done. In order to improve the paper further, the figures should be simplified (see below).

Comments

1. Title is a bit too generic/vague and needs to be changed. What about "Cryo-electron microscopy reveals that the *Methanocaldococcus villosus* archaellum filament is a heteropolymer composed of two alternating subunits"?

2. Lines 35-36: sentence is vague. It should be clearly stated what the "two major structural elements that enable the archaellum filament to move" are.

3. Lines 117-118: for those who do not know what an Ultraturrax T25 is, make it clear how the shearing of the filaments is done.

4. Most of the Figures are really crowded and could be improved by keeping only the absolutely essential/most meaningful panels (transferring the others to Supplementary data). This is my main criticism of the paper.

Fig. 1 should only consist of panels f, g, e and d. It would be good to highlight the helical symmetry parameters on the figure.

Fig. 2: do we really need four views for each subunit in panel e? Same applies to panel f.

Fig. 3: panels b and c are not really essential since most of the density information is conveyed in panels d and e.

Fig. 4: panels d, e, f and h are not really essential. It would be good to highlight with a close-up the interactions between ArlB1 and ArlB2 on this figure.

5. Lines 270-271: why is ArlB3 deemed "too long to fit into either position"? According to Suppl. fig. 7, all three subunits have similar sizes.

6. Line 304: why do the author write "We were able to fit four of the seven sugar units into our map" since Fig. 3 show sugar fitting in all seven glycosylation sites?

7. Lines 377-379: comparison with bacterial T4P is unwarranted because there is no hinge in bacterial pilins where the globular head is packed against the end of the N-terminal α -helix.

8. Throughout the manuscript replace "ArlB1 and 2" by "ArlB1 and ArlB2".

9. Line 399: "arlB1- and arlB2- mutants" is not standard genetic nomenclature. It should be " Δ arlB1 and Δ arlB2 mutants".

Minor comments

1. Line 51: "TFF filament" should be "T4F". Also, to make it consistent with the T4P abbreviation,

type IV filament should be abbreviated as T4F.

2. Line 52: the role of T4P electrical conductance is questionable according to many recent studies.

3. Line 54: delete "subunits", a term which should be restricted to pilins.

4. Line 105: "trace mineral solution 10-fold" should be "10X trace mineral solution".

5. Line 145: there is a typo in "archaeallum".

6. Line 217: "proteinA" should be "protein A".

7. Suppl. fig. 11: the colours in panel a are too dark to easily distinguish them. They should be brighter.

8. Suppl. fig. 14: in panel b, replace "unconserved" by "variable".

9. Line 403: "considerably" is an overstatement.

10. Line 414: define "ecoparalogues".

11. Line 513: what does "indiscrete manner" mean?

Reviewer #2 (Remarks to the Author):

This manuscript presents the first high resolution structure of the archaeallum filament composed of two alternating archaeellins, in contrast to previous reported homopolymeric archaeallum filaments . The structure was determine using cryo-EM and helical reconstruction. This is a very solid structure and presented in an easy to follow and well written manuscript.

The authors identified the N-glycosylation sites though which sugar is not identified. The authors go on to apply 3DVA analysis to show the flexibility of the structure and put insight on the mechanism of bending or rotation of the filaments. This is very beautiful work and I have only a few minor comments.

Comments:

1. The authors cite three high-resolution structures of archaeella that are available to date show homopolymeric filaments consisting of only single archaeellin. It could be either only one major archaeellin in these systems or inappropriate image processing, as the authors showed if a wrong symmetry applied. So it is better to clarify in the background statement.

2. ArlB1 and 2 could each be modelled unambiguously into alternating positions in the map, as the authors stated the model building was guided by aromatic side chains and glycosylation sites. I do not doubt on this, the authors shown the different glycosylation sites of ArlB1 and 2, could the author show some examples of side chains fitting which could help to distinguish ArlB1 and 2 ?

3. In the Suppl. fig. 5, in my understanding, the golden color represents the resolution around 3.08 Å, is that right?

Reviewer #3 (Remarks to the Author):

The author showed a high-quality structure of the archaeella, which show complicated helical

symmetry. The careful analysis of helical structure show a new architecture of archaella with two components, ArlB1 and 2. The 3DVA by cryoSparc was used to detect the structural flexibility. The overall structure has high-resolution and high-quality.

several concerns:

- 1) line 134: CTF-corrected using CTF-find 4. CTF-find 4 is usually written in CTFFind 4, which cannot carry out CTF correction, should be CTF estimation.
- 2) ArlB1 and 2 were distinguished by the differences in densities. The comparison between two maps should be shown in details, such as different side-chain densities, to demonstrate the differences and the unambiguity in model building.
- 3) While the 3DVA generated many map frames, the flexibility is very subtle. Because the flexibility exists for nearly all soft protein macromolecules, 3DVA always produces some results, either functional relevant or not. Therefore, correlating such flexibility to the functional filament motion is risky, further biological validation is needed.

Point-by-point response to the reviewers' comments

Reviewer 1

Summary

The manuscript entitled "New insights into the architecture and dynamics of archaella" by Gambelli et al. focuses on the archaellum in Methanocaldococcus villosus. Archaella are swimming filaments, functional analogs of bacterial flagella but belonging to the superfamily of type 4 filaments (T4F). In this paper, the authors present the structure of the M. villosus archaellum determined by cryoEM at 3.08 Å resolution. This is the first heteropolymeric structure of a T4F since it consists of two alternating subunits, ArlB1 and ArlB2. There is evidence that this heteropolymeric organisation might be quite frequent in the T4F superfamily. They propose a very interesting model in which an ArlB1-ArlB2 heterodimer, that exhibits extensive intermolecular interactions, would be the building block of the filament. The paper is well written and the experiments are carefully done. In order to improve the paper further, the figures should be simplified (see below).

Comments

1. Title is a bit too generic/vague and needs to be changed. What about "Cryo-electron microscopy reveals that the Methanocaldococcus villosus archaellum filament is a heteropolymer composed of two alternating subunits"?

We adjusted the title according to reviewer's comment but left out "a heteropolymer", to limit the number of words in the new title. The new title is:

"Electron cryo-microscopy reveals that the Methanocaldococcus villosus archaellum filament is composed of two alternating subunits"

2. Lines 35-36: sentence is vague. It should be clearly stated what the "two major structural elements that enable the archaellum filament to move" are.

Adjusted according to reviewer's comment. See line 39-40 of track changes document. The re-worked abstract now includes the following statement:

"Moreover, we identify two structural elements that determine the filament's flexibility – intramolecular hinges and intermolecular grease."

3. Lines 117-118: for those who do not know what an Ultraturrax T25 is, make it clear how the shearing of the filaments is done.

Clarified according to reviewer's comment. See line 413-417 of track changes document.
We have explained in more detail the shearing process.

4. Most of the Figures are really crowded and could be improved by keeping only the absolutely essential/most meaningful panels (transferring the others to Supplementary data). This is my main criticism of the paper.

We addressed the reviewer's comment by simplifying the figures

Fig. 1 should only consist of panels f, g, e and d. It would be good to highlight the helical symmetry parameters on the figure.

We have removed panels a, b and c and moved them to suppl. fig. 4. We have also added the helical symmetry parameters to panel a.

Fig. 2: do we really need four views for each subunit in panel e? Same applies to panel f.

We have removed two views each from panels d, e and f.

Fig. 3: panels b and c are not really essential since most of the density information is conveyed in panels d and e.

We have removed panels b and c.

Fig. 4: panels d, e, f and h are not really essential. It would be good to highlight with a close-up the interactions between ArlB1 and ArlB2 on this figure.

We have simplified the figure by removing panels d, e and f, which are now in suppl. fig. 14. Two close ups of the interactions between ArlB1 and ArlB2 are now in panel d.

5. Lines 270-271: why is ArlB3 deemed "too long to fit into either position"? According to Suppl. fig. 7, all three subunits have similar sizes.

ArlB3 is only slightly longer than ArlB1 (14 amino acids longer) and ArlB2 (10 amino acids longer), and we understand this sentence may be confusing for the reader, so we have removed it from the text. See lines 118-119 of track changes document.

6. Line 304: why do the author write “We were able to fit four of the seven sugar units into our map” since Fig. 3 show sugar fitting in all seven glycosylation sites?

We could fit atomic models for glycans in all sugar densities found in our map. However, for each glycan density, we could only see polysaccharides consisting of four sugars. We changed the wording to avoid any confusion. See lines 153-157 of the track changes document:

“For each of the glycan densities, we were able to fit only the first four sugars of the *M. thermolithotrophicus* polysaccharide. No additional saccharides were resolved beyond the fourth unit, which is likely due to their flexibility or a shorter glycan in *M. villosus*.”

7. Lines 377-379: comparison with bacterial T4P is unwarranted because there is no hinge in bacterial pilins where the globular head is packed against the end of the N-terminal α -helix.

We concur with the reviewer’s comment and adapted our Discussion (lines 353-365 in the tracked document):

“The amino acids that make up the loop of non-motile archaeal pili contain larger side chains (QQT/QVT/QGT) than those of archaella (SG). This may infer more steric hindrance and less flexibility of pili-borne compared to archaellar hinges. Based on this, the archaeal hinge region could be used to distinguish between archaellins and archaeal type-4 pilins when predicting their function at the sequence level. In bacterial type-4 pilins, the globular head is packed against the N-terminal α -helix, likely resulting in further decreased flexibility of this area compared to archaellins or archaeal pilins. Instead, bacterial type-4 pilins contain a conserved “melted” region within the N-terminal α -helix, which could aid their flexibility in a similar fashion.”

8. Throughout the manuscript replace “ArlB1 and 2” by “ArlB1 and ArlB2”.

Adjusted according to reviewer's comment throughout the track changes document.

9. Line 399: "*arlB1- and arlB2- mutants*" is not standard genetic nomenclature. It should be "*ΔarlB1 and ΔarlB2 mutants*".

Amended according to reviewer's comment. See lines 258 and 261 of track changes document.

Minor comments

1. Line 51: "*TFF filament*" should be "*TFF*". Also, to make it consistent with the T4P abbreviation, *type IV filament* should be abbreviated as T4F.

Adjusted according to reviewer's comment. See line 52 and throughout the text of the track changes document.

2. Line 52: *the role of T4P electrical conductance is questionable according to many recent studies.*

We agree with the reviewer and have removed this statement from the manuscript.

3. Line 54: delete "*subunits*", a term which should be restricted to *pilins*.

Amended according to reviewer's comment. See line 58 of track changes document.

4. Line 105: "*trace mineral solution 10-fold*" should be "*10X trace mineral solution*".

Adjusted according to reviewer's comment. See line 400 of track changes document.

5. Line 145: there is a typo in "*archaeallum*".

Amended according to reviewer's comment. See line 445 of track changes document.

6. Line 217: "*proteinA*" should be "*protein A*".

Edited according to reviewer's comment. See line 517 of track changes document.

7. *Suppl. fig. 11: the colours in panel a are too dark to easily distinguish them. They should be brighter.*

We increased the brightness of the figure. See new suppl. fig. 13.

8. *Suppl. fig. 14: in panel b, replace "unconserved" by "variable".*

The figure panel b now says "variable" instead of unconserved. See new suppl. fig. 17.

9. *Line 403: "considerably" is an overstatement.*

Adjusted according to reviewer's comment. To moderate the statement, we have removed "considerably" from the sentence. See line 262 of track changes document.

10. *Line 414: define "ecoparalogues".*

We defined the word in the revised manuscript. See line 270-272 of track changes document:

"Furthermore, in *Haloarcula marismortui* the two archaeellins ArIA2 and ArIB are ecoparalogues, meaning that different archaeella are assembled in response to varying environmental situations (in this case the level of salinity in the environment)."

11. *Line 513: what does "indiscrete manner" mean?*

We changed the wording to clarify the reviewer's comment. See line 381 of track changes document:

"In contrast, our data suggest that archaeellins change their shape in a continuous manner."

Reviewer 2

Summary

This manuscript presents the first high resolution structure of the archaellum filament composed of two alternating archaellins, in contrast to previous reported homopolymeric archaellum filaments . The structure was determine using cryo-EM and helical reconstruction. This is a very solid structure and presented in an easy to follow and well written manuscript.

The authors identified the N-glycosylation sites though which sugar is not identified. The authors go on to apply 3DVA analysis to show the flexibility of the structure and put insight on the mechanism of bending or rotation of the filaments. This is very beautiful work and I have only a few minor comments.

Comments

1. The authors cite three high-resolution structures of archaella that are available to date show homopolymeric filaments consisting of only single archaellin. It could be either only one major archaellin in these systems or inappropriate image processing, as the authors showed if a wrong symmetry applied. So it is better to clarify in the background statement.

We agree with the reviewer's comment. The previously published structures could either be *bona fide* homopolymers or heteropolymers that were overlooked by applying symmetry parameters that averaged these heterogeneities out. We have revisited our own previously published filament from *Pyrococcus furiosus* using our new image processing procedure but at the obtained resolution of 4.5 Å, the new structure does not reveal any noticeable heterogeneities that could account for alternating subunits. As to the other two published structures, these have been published by different groups and we will not know the answer unless those groups reprocess their data. Without having ultimate proof, we would prefer to be cautious with any judgements and therefore included the following statement in the discussion:

Line 249-252: "In this regard, it would be interesting to revisit previously published structures of archaella and use our image processing strategy to investigate if these are indeed homopolymers by nature or perhaps also consist of two alternating subunits."

2. ArlB1 and 2 could each be modelled unambiguously into alternating positions in the map, as the authors stated the model building was guided by aromatic side chains and glycosylation sites. I do not doubt on this, the authors shown the different glycosylation sites of ArlB1 and 2, could the author show some examples of side chains fitting which could help to distinguish ArlB1 and 2?

We addressed the reviewers comment and prepared a new supplementary figure 7, clearly showing the key differences between the maps and models for ArlB1 and ArlB2. See new suppl. fig. 7.

3. In the Suppl. fig. 5, in my understanding, the golden color represents the resolution around 3.08 Å, is that right?

Yes, that is correct.

Reviewer 3

Summary

The author showed a high-quality structure of the archaella, which show complicated helical symmetry. The careful analysis of helical structure show a new architecture of archaella with two components, ArlB1 and 2. The 3DVA by cryoSparc was used to detect the structural flexibility. The overall structure has high-resolution and high-quality. several concerns:

Comments

1) line 134: CTF-corrected using CTF-find 4. CTF-find 4 is usually written in CTFFind 4, which cannot carry out CTF correction, should be CTF estimation.

Adjusted according to reviewer's comment. See line 437-438 of track changes document:

"Briefly, the movies were motion-corrected using MotionCorr 2⁶⁸ and CTF estimation was carried out using CTFFind 4"

2) ArlB1 and 2 were distinguished by the differences in densities. The comparison between two maps should be shown in details, such as different side-chain densities, to demonstrate the differences and the unambiguity in model building.

We addressed the reviewer's comment by preparing a new supplementary figure 7, clearly showing the key differences between the maps and models for ArlB1 and ArlB2. See new suppl. fig. 7.

3) While the 3DVA generated many map frames, the flexibility is very subtle. Because the flexibility exists for nearly all soft protein macromolecules, 3DVA always produces some results, either functional relevant or not. Therefore, correlating such flexibility to the functional filament motion is risky, further biological validation is needed.

We have addressed the reviewer's comment. See line 373-375 of track changes document:

"It is important to point out that the flexibility we observe is subtle, and that further biological validation is needed to undoubtedly link such flexibility to the functional filament motion."